# A descriptive analysis of the data availability statements accompanying medRxiv preprints and a comparison with their published counterparts

**Luke A. McGuinness**[1,2]*, **Athena L. Sheppard**[3]

**1** Population Health Sciences, Bristol Medical School, University of Bristol, Bristol, United Kingdom, **2** MRC Integrative Epidemiology Unit at the University of Bristol, Bristol, United Kingdom, **3** Department of Health Sciences, University of Leicester, Leicester, United Kingdom

* luke.mcguinness@bristol.ac.uk

**Data Availability Statement:** All materials (data, code and supporting information) are available from GitHub (https://github.com/mcguinlu/data-

## Abstract

### Objective

To determine whether medRxiv data availability statements describe open or closed data—that is, whether the data used in the study is openly available without restriction—and to examine if this changes on publication based on journal data-sharing policy. Additionally, to examine whether data availability statements are sufficient to capture code availability declarations.

### Design

Observational study, following a pre-registered protocol, of preprints posted on the medRxiv repository between 25th June 2019 and 1st May 2020 and their published counterparts.

### Main outcome measures

Distribution of preprinted data availability statements across nine categories, determined by a prespecified classification system. Change in the percentage of data availability statements describing open data between the preprinted and published versions of the same record, stratified by journal sharing policy. Number of code availability declarations reported in the full-text preprint which were not captured in the corresponding data availability statement.

### Results

3938 medRxiv preprints with an applicable data availability statement were included in our sample, of which 911 (23.1%) were categorized as describing open data. 379 (9.6%) preprints were subsequently published, and of these published articles, only 155 contained an applicable data availability statement. Similar to the preprint stage, a minority (59 (38.1%)) of these published data availability statements described open data. Of the 151 records eligible for the comparison between preprinted and published stages, 57 (37.7%) were

availability-impact), archived at time of submission on Zenodo (DOI: 10.5281/zenodo.3968301).

**Funding:** LAM is supported by an National Institute for Health Research (NIHR; https://www.nihr.ac.uk/) Doctoral Research Fellowship (DRF-2018-11-ST2-048). The funders had no role in study design, data collection and analysis, decision to publish, or preparation of the manuscript. The views expressed in this article are those of the authors and do not necessarily represent those of the NHS, the NIHR, MRC, or the Department of Health and Social Care.

**Competing interests:** The authors have declared that no competing interests exist.

published in journals which mandated open data sharing. Data availability statements more frequently described open data on publication when the journal mandated data sharing (open at preprint: 33.3%, open at publication: 61.4%) compared to when the journal did not mandate data sharing (open at preprint: 20.2%, open at publication: 22.3%).

## Conclusion

Requiring that authors submit a data availability statement is a good first step, but is insufficient to ensure data availability. Strict editorial policies that mandate data sharing (where appropriate) as a condition of publication appear to be effective in making research data available. We would strongly encourage all journal editors to examine whether their data availability policies are sufficiently stringent and consistently enforced.

## 1 Introduction

The sharing of data generated by a study is becoming an increasingly important aspect of scientific research [1, 2]. Without access to the data, it is harder for other researchers to examine, verify and build on the results of that study [3]. As a result, many journals now mandate data availability statements. These are dedicated sections of research articles, which are intended to provide readers with important information about whether the data described by the study are available and if so, where they can be obtained [4].

While requiring data availability statements is an admirable first step for journals to take, and as such is viewed favorably by journal evaluation rubrics such as the Transparency and Openness Promotion [TOP] Guidelines [5], a lack of review of the contents of these statements often leads to issues. Many authors claim that their data can be made "available on request", despite previous work establishing that these statements are demonstrably untrue in the majority of cases—that when data is requested, it is not actually made available [6–8]. Additionally, previous work found that the availability of data "available on request" declines with article age, indicating that this approach is not a valid long term option for data sharing [9]. This suggests that requiring data availability statements without a corresponding editorial or peer review of their contents, in line with a strictly enforced data-sharing policy, does not achieve the intended aim of making research data more openly available. However, few journals actually mandate data sharing as a condition of publication. Of a sample of 318 biomedical journals, only ~20% had a data-sharing policy that mandated data sharing [10].

Several previous studies have examined the data availability statements of published articles [4, 11–13], but to date, none have examined the statements accompanying preprinted manuscripts, including those hosted on medRxiv, the preprint repository for manuscripts in the medical, clinical, and related health sciences [14]. Given that preprints, particularly those on medRxiv, have impacted the academic discourse around the recent (and ongoing) COVID-19 pandemic to a similar, if not greater, extent than published manuscripts [15], assessing whether these studies make their underlying data available without restriction (i.e. "open"), and adequately describe how to access it in their data availability statements, is worthwhile. In addition, by comparing the preprint and published versions of the data availability statements for the same paper, the potential impact of different journal data-sharing policies on data availability can be examined. This study aimed to explore the distribution of data availability statements' description of the underlying data across a number of categories of "openness" and to

assess the change between preprint and journal-published data availability statements, stratified by journal data-sharing policy. We also intended to examine whether authors planning to make the data available upon publication actually do so, and whether data availability statements are sufficient to capture code availability declarations.

## 2 Methods

### 2.1 Protocol and ethics

A protocol for this analysis was registered in advance and followed at all stages of the study [16]. Any deviations from the protocol are described. Ethical approval was not required for this study.

### 2.2 Data extraction

The data availability statements of preprints posted on the medRxiv preprint repository between 25th June 2019 (the date of first publication of a preprint on medRxiv) and 1st May 2020 were extracted using the medrxivr and rvest R packages [17, 18]. Completing a data availability statement is required as part of the medRxiv submission process, and so a statement was available for all eligible preprints. Information on the journal in which preprints were subsequently published was extracted using the published DOI provided by medRxiv and rcrossref [19]. Several other R packages were used for data cleaning and analysis [20–33].

To extract the data availability statements for published articles and the journals data-sharing policies, we browsed to the article or publication website and manually copied the relevant material (where available) into an Excel file. The extracted data are available for inspection (see Material availability section).

### 2.3 Categorization

A pre-specified classification system was developed to categorize each data availability statement as describing either open or closed data, with additional ordered sub-categories indicating the degree of openness (see Table 1). The system was based on the "Findability" and "Accessibility" elements of the FAIR framework [34], the categories used by previous effort to categorize published data availability statements [4, 11], our own experience of medRxiv data availability statements, and discussion with colleagues. Illustrative examples of each category were taken from preprints included in our sample [35–43].

The data availability statement for each preprinted record were categorized by two independent researchers, using the groups presented in Table 1, while the statements for published articles were categorized using all groups barring Category 3 and 4 ("Available in the future"). Records for which the data availability statement was categorized as "Not applicable" (Category 1 from Table 1) at either the preprint or published stage were excluded from further analyses. Researchers were provided only with the data availability statement, and as a result, were blind to the associated preprint metadata (e.g. title, authors, corresponding author institution) in case this could affect their assessments. Any disagreements were resolved through discussion.

Due to our large sample, if authors claimed that all data were available in the manuscript or as a S1 File, or that their study did not make use of any data, we took them at their word. Where a data availability statement met multiple categories or contained multiple data sources with varying levels of openness, we took a conservative approach and categorized it on the basis of the most restrictive aspect (see S1 File for some illustrative examples). We plotted the

**Table 1. Categories used to classify the data availability statements.**

| Key | Main category | Sub-category | Example |
|---|---|---|---|
| 0 | Not applicable (protocol for a review, commentary, etc) | | "Data sharing not applicable to this article as no datasets were generated or analysed during the current study." [35] |
| 1 | "Closed" | Data not made available | "Not available for public" [36] |
| 2 | "Closed" | Data available on request to authors | "Data can be available upon reasonable request to the corresponding author." [37] |
| 3 | "Closed" | Data will be made available in the future (link provided) | "The protocol and full dataset will be available at Open Science Framework upon peer review publication (https://osf.io/rvbuy/)." [38] |
| 4 | "Closed" | Data will be made available in the future (no link provided) | "Data will be deposited in Dryad upon publication" [39] |
| 5 | "Closed" | Data available from central repository (access-controlled or open access), but insufficient detail available to find specific dataset | "Data were obtained from the international MSBase cohort study. Information regarding data availability can be obtained at https://www.msbase.org/." OR Daily diagnosis number of countries outside China is download from WHO situation reports (https://www.who.int/emergencies/diseases/novel-coronavirus-2019/situation-reports). https://www.who.int/emergencies/diseases/novel-coronavirus-2019/situation-reports [40] |
| 6 | "Closed" | Data available from central access-controlled repository, and sufficient details included to identify specific dataset e.g. via extract or accession ID or date stamp | "This research has been conducted using the UK Biobank Resource under application number 24494. All bona fide researchers can apply to use the UK Biobank resource for health related research that is in the public interest." [41] |
| 7 | "Open" | Data available in the manuscript/S1 File | "All data related to this study are present in the paper or the S1 File." [42] |
| 8 | "Open" | Data available via a online repository that is not access-controlled e.g. Dryad, Zenodo | "Extracted data used in this meta-analysis and analysis code are available at www.doi.org/10.5281/zenodo.3149365." [43] |

Illustrative examples of each category were taken from preprints included in our sample (see "Data extraction").

distribution of preprint and published data availability statements across the nine categories presented in Table 1.

Similarly, the extracted data-sharing policies were classified by two independent reviewers according to whether the journal mandated data sharing (1) or not (0). Where the journal had no obvious data sharing policy, these were classified as not mandating data sharing.

## 2.4 Changes between preprinted and published statements

To assess if data availability statements change between preprint and published articles, we examined whether a discrepancy existed between the categories assigned to the preprinted and published statements, and the direction of the discrepancy ("more closed" or "more open"). Records were deemed to become "more open" if their data availability statement was categorized as "closed" at the preprint stage and "open" at the published stage. Conversely, records described as "more closed" were those moving from "open" at preprint to "closed" on publication.

We declare a minor deviation from our protocol for this analysis [16]. Rather than investigating the data-sharing policy only for journals with the largest change in openness as intended, which involved setting an arbitrary cut-off when defining "largest change", we systematically extracted and categorized the data-sharing policies for all journals in which preprints had subsequently been published using two categories (1: "requiring/mandating data sharing" and, 2: "not requiring/mandating data sharing"), and compared the change in openness between these two categories. Note that Category 2 includes journals that encourage data sharing, but do not make it a condition of publication.

To assess claims that data will be provided on publication, the data availability statements accompanying the published articles for all records in Category 3 ("Data available on publication (link provided)") or Category 4 ("Data available on publication (no link provided)") from Table 1 were assessed, and any difference between the two categories examined.

## 2.5 Code availability

Finally, to assess whether data availability statements also capture the availability of programming code, such as STATA do files or R scripts, the data availability statement and full text PDF for a random sample of 400 preprinted records were assessed for code availability (1: "code availability described" and 2: "code availability not described").

## 3 Results

The data availability statements accompanying 4101 preprints registered between 25th June 2019 and 1st May 2020 were extracted from the medRxiv preprint repository on the 26th May 2020 and were coded by two independent researchers according to the categories in Table 1. During this process, agreement between the raters was high (Cohen's Kappa = 0.98; "almost perfect agreement") [44].

Of the 4101 preprints, 163 (4.0%) in Category 0 ("Not applicable") were excluded following coding, leaving 3938 remaining records. Of these, 911 (23.1%) had made their data open as per the criteria in Table 1. The distribution of data availability statements across the categories can be seen in Fig 1. A total of 379 (9.6%) preprints had been subsequently published, and of these, only 159 (42.0%) had data availability statements that we could categorize. 4 (2.5%) records in

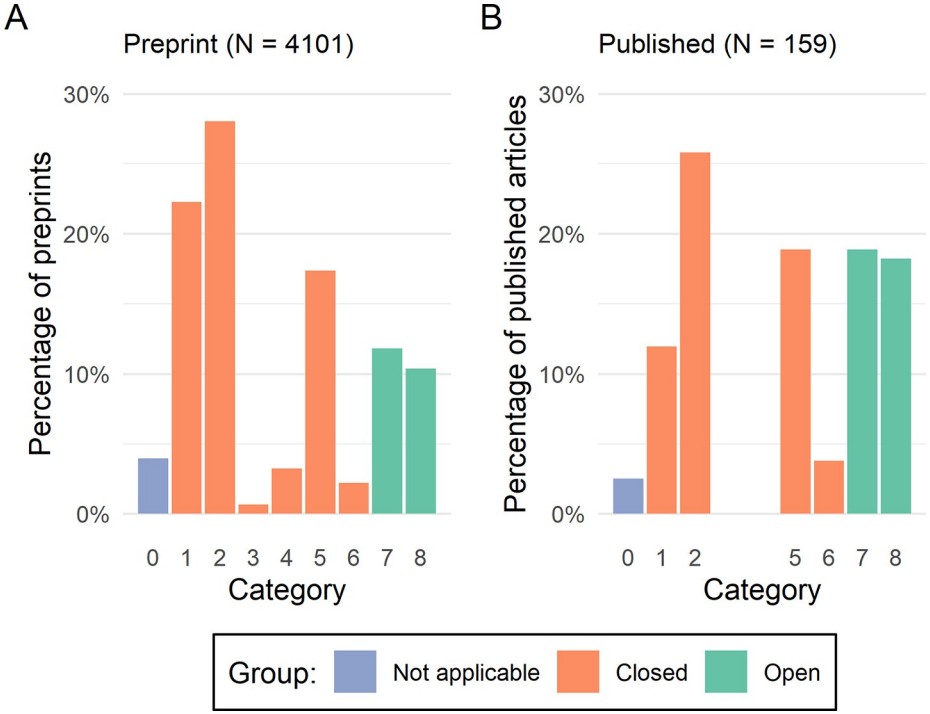

**Fig 1. Distribution of the data availability statements of preprinted (Panel A) and published (Panel B) records by category from Table 1.**

**Table 2. Change in openness of data availability statements from preprint to published article, grouped by journal data-sharing policy.**

| Journal data sharing policy | Preprinted records subsequently published (N) | Open DAS in preprinted version % (N) | Open DAS in published version % (N) | Change in DAS from preprint to publication | | |
| --- | --- | --- | --- | --- | --- | --- |
| | | | | More open (N) | More closed (N) | No change (N) |
| **Does not mandate open data** | 94 | 20.2% (19) | 22.3% (21) | 10 | 8 | 76 |
| **Mandates open data** | 57 | 33.3% (19) | 61.4% (35) | 16 | 0 | 41 |

Category 0 ("Not applicable") were excluded, and of the 155 remaining, 59 (38.1%) had made their data open as per our criteria.

For the comparison of preprinted data availability statements with their published counterparts, we excluded records that were not published, that did not have a published data availability statement or that were labeled as "Not applicable" at either the preprint or published stage, leaving 151 records (3.7% of the total sample of 4101 records) records.

Data availability statements more frequently described open data on publication compared to the preprinted record when the journal mandated data sharing (Table 2). Moreover, the data availability statements for 8 articles published in journals that did not mandate open data sharing became less open on publication. The change in openness for preprints grouped by category and stratified by journal policy is shown in S1 Table in S1 File, while the change for each individual journal included in our analysis is shown in S2 Table in S1 File.

Interestingly, 22 records published in a journal mandating open data sharing did not have an open data availability statement. The majority of these records described data that was available from a central access-controlled repository (Category 5 or 6), while in others, legal restrictions were cited as the reason for lack of data sharing. However, in some cases, data was either insufficiently described or was only available on request (S3 Table in S1 File), indicating that journal policies which mandate data sharing may not always be consistently applied allowing some records may slip through the gaps.

161 (4.1%) preprints stated that data would be available on publication, but only 10 of these had subsequently been published (Table 3) and the number describing open data on publication did not seem to vary based on whether the preprinted data availability statements include a link to an embargoed repository or not, though the sample size is small.

Of the 400 records for which code availability was assessed, 75 mentioned code availability in the preprinted full-text manuscript. However, only 22 (29.3%) of these also described code availability in the corresponding data availability statement (S4 Table in S1 File).

**Table 3. Assessment of whether researchers promising to make data available on publication actually do so, and whether this differs if researchers included a link to an embargoed repository or not.**

| Preprint Category | Number of preprints | Published Category | Number of published studies |
| --- | --- | --- | --- |
| **Data available in the future, with a link to an embargoed repository provided** | 3 | 1. Data not made available | 1 (33.3%) |
| | | 5. Data available from central repository (access-controlled or open access), but insufficient detail available to find specific dataset | 1 (33.3%) |
| | | 8. Data available via a online repository that is not access-controlled e.g. Dryad, Zenodo | 1 (33.3%) |
| **Data available in the future, with no details of embargoed repository given** | 7 | 1. Data not made available | 1 (14.3%) |
| | | 2. Data available on request to authors | 1 (14.3%) |
| | | 7. Data available in the manuscript/S1 File | 1 (14.3%) |
| | | 8. Data available via a online repository that is not access-controlled e.g. Dryad, Zenodo | 4 (57.1%) |

# 4 Discussion

## 4.1 Principal findings and comparison with other studies

We have reviewed 4101 preprinted and 159 published data availability statements, coding them as "open" or "closed" according to a predefined classification system. During this labor-intensive process, we appreciated statements that reflected the authors' enthusiasm for data sharing ("YES") [45], their bluntness ("Data is not available on request.") [46], and their efforts to endear themselves to the reader ("I promise all data referred to in the manuscript are available.") [47]. Of the preprinted statements, almost three-quarters were categorized as "closed", with the largest individual category being "available on request". In light of the substantial impact that studies published as preprints on medRxiv have had on real-time decision making during the current COVID-19 pandemic [15], it is concerning that data for these preprints is so infrequently readily available for inspection.

A minority of published records we examined contained a data availability statement (n = 159 (42.0%)). This lack of availability statement at publication results in a loss of useful information. For at least one published article, we identified relevant information in the preprinted statement that did not appear anywhere in the published article, due to it not containing a data availability statement [48, 49].

We provide initial descriptive evidence that strict data-sharing policies, which mandate that data be made openly available (where appropriate) as a condition of publication, appear to succeed in making research data more open than those that do not. Our findings, though based on a relatively small number of observations, agree with other studies on the effect of journal policies on author behavior. Recent work has shown that "requiring" a data availability statement was effective in ensuring that this element was completed [4], while "encouraging" authors to follow a reporting checklist (the ARRIVE checklist) had no effect on compliance [50, 51].

Finally, we also provide evidence that data availability statements alone are insufficient to capture code availability declarations. Even when researchers wish to share their code, as evidenced by a description of code availability in the main paper, they frequently do not include this information in the data availability statement. Code sharing has been advocated strongly elsewhere [52–54], as it provides an insight into the analytic decisions made by the research team, and there are few, if any, circumstances in which it is not possible to share the analytic code underpinning an analysis. Similar to data availability statements, a dedicated code availability statement which is critically assessed against a clear code-sharing policy as part of the editorial and peer review processes will help researchers to appraise published results.

## 4.2 Strengths and limitations

A particular strength of this analysis is that the design allows us to compare what is essentially the same paper (same design, findings and authorship team) under two different data-sharing polices, and assess the change in the openness of the statement between them. To our knowledge this is the first study to use this approach to examine the potential impact of journal editorial policies. This approach also allows us to address the issue of self-selection. When looking at published articles alone, it is not possible to tell whether authors always intended to make their data available and chose a given journal due to its reputation for data sharing. In addition, we have examined all available preprints within our study period and all corresponding published articles, rather than taking a sub-sample. Finally, categorization of the statements was carried out by two independent researchers using predefined categories, reducing the risk of misclassification.

However, our analysis is subject to a number of potential limitations. The primary one is that manuscripts (at both the preprint and published stages) may have included links to the data, or more information that uniquely identifies the dataset from a data portal, within the text (for example, in the Methods section). While this might be the case, if readers are expected to piece together the relevant information from different locations in the manuscript, it throws into question what having a dedicated data availability statement adds. A second limitation is that we do not assess the veracity of any data availability statements, which may introduce some misclassification bias into our categorization. For example, we do not check whether all relevant data can actually be found in the manuscript/S1 File (Category 7) or the linked repository (Category 8), meaning our results provide a conservative estimate of the scale of the issue, asprevious work has suggested that this is unlikely to be the case [12]. A further consideration is that for Categories 1 ("No data available") and 2 ("Available on request"), there will be situations where making research data available is not feasible, for example, due to cost or concerns about patient re-identifiability [55, 56]. This situation is perfectly reasonable, as long as statements are explicit in justifying the lack of open data.

## 4.3 Implications for policy

Data availability statements are an important tool in the fight to make studies more reproducible. However, without critical review of these statements in line with strict data-sharing policies, authors default to not sharing their data or making it "available on request". Based on our analysis, there is a greater change towards describing open data between preprinted and published data availability statements in journals that mandate data sharing as a condition of publication. This would suggest that data sharing could be immediately improved by journals becoming more stringent in their data availability policies. Similarly, introduction of a related code availability section (or composite "material" availability section) will aid in reproducibility by capturing whether analytic code is available in a standardized manuscript section.

It would be unfair to expect all editors and reviewers to be able to effectively review the code and data provided with a submission. As proposed elsewhere [57], a possible solution is to assign an editor or reviewer whose sole responsibility in the review process is to examine the data and code provided. They would also be responsible for judging, when data and code are absent, whether the argument presented by the authors for not sharing these materials is valid.

However, while this study focuses primarily on the role of journals, some responsibility for enacting change rests with the research community at large. If researchers regularly shared our data, strict journal data-sharing policies would not be needed. As such, we would encourage authors to consider sharing the data underlying future publications, regardless of whether the journal actually mandates it.

## 5 Conclusion

Requiring that authors submit a data availability statement is a good first step, but is insufficient to ensure data availability, as our work shows that authors most commonly use them to state that data is only available on request. However, strict editorial policies that mandate data sharing (where appropriate) as a condition of publication appear to be effective in making research data available. In addition to the introduction of a dedicated code availability statement, a move towards mandated data sharing will help to ensure that future research is readily reproducible. We would strongly encourage all journal editors to examine whether their data availability policies are sufficiently stringent and consistently enforced.

## Supporting information

**S1 File.**
(DOCX)

## Acknowledgments

We must acknowledge the input of several people, without whom the quality of this work would have been diminished: Matthew Grainger and Neal Haddaway for their insightful comments on the subject of data availability statements; Phil Gooch and Sarah Nevitt for their skill in identifying missing published papers based on the vaguest of descriptions; Antica Culina, Julian Higgins and Alfredo Sánchez-Tójar for their comments on the preprinted version of this article; and Ciara Gardiner, for proof-reading this manuscript.

## Author Contributions

**Conceptualization:** Luke A. McGuinness.

**Data curation:** Luke A. McGuinness.

**Formal analysis:** Luke A. McGuinness, Athena L. Sheppard.

**Investigation:** Luke A. McGuinness, Athena L. Sheppard.

**Methodology:** Luke A. McGuinness, Athena L. Sheppard.

**Project administration:** Luke A. McGuinness.

**Software:** Luke A. McGuinness.

**Supervision:** Luke A. McGuinness.

**Validation:** Luke A. McGuinness, Athena L. Sheppard.

**Visualization:** Luke A. McGuinness.

**Writing – original draft:** Luke A. McGuinness.

**Writing – review & editing:** Luke A. McGuinness, Athena L. Sheppard.

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
