## [Decision Letter · Decision Letter 0]

3 Feb 2021

PONE-D-20-29718

A descriptive analysis of the data availability statements accompanying medRxiv preprints and a comparison with their published counterparts

PLOS ONE

Dear Dr. McGuinness,

Thank you for submitting your manuscript to PLOS ONE. After careful consideration, we feel that it has merit but does not fully meet PLOS ONE’s publication criteria as it currently stands. Therefore, we invite you to submit a revised version of the manuscript that addresses the points raised during the review process.

The reviewers suggested minor revisions. Please, reviewed that carefully.

We look forward to receiving your revised manuscript.

Kind regards,

Rafael Sarkis-Onofre

Academic Editor

PLOS ONE

Journal Requirements:

Reviewers' comments:

Reviewer's Responses to Questions

**Comments to the Author**

1. Is the manuscript technically sound, and do the data support the conclusions?

Reviewer #1: Yes

Reviewer #2: Yes

2. Has the statistical analysis been performed appropriately and rigorously? 

Reviewer #1: Yes

Reviewer #2: Yes

3. Have the authors made all data underlying the findings in their manuscript fully available?

Reviewer #1: Yes

Reviewer #2: Yes

4. Is the manuscript presented in an intelligible fashion and written in standard English?

Reviewer #1: Yes

Reviewer #2: Yes

5. Review Comments to the Author

Reviewer #1: 1. The study presents the results of original research.

Yes.

2. Results reported have not been published elsewhere.

Yes, the authors state that the results were not published elsewhere.

3. Experiments, statistics, and other analyses are performed to a high technical standard and are described in sufficient detail.

Only a descriptive analysis was performed, but detailed described.

4. Conclusions are presented in an appropriate fashion and are supported by the data.

Yes.

5. The article is presented in an intelligible fashion and is written in standard English.

Yes.

6. The research meets all applicable standards for the ethics of experimentation and research integrity.

Yes.

7. The article adheres to appropriate reporting guidelines and community standards for data availability.

Yes.

General comments

The study idea is original and reinforce the science path in direction to the transparency in research.

The methodological part description should be improved to facilitate the understanding.

The presentation and description of the results could be improved for a better understanding, including the tables.

I would like to suggest the replacement of the terms “more open” and “more closed” for more suitable terms.

The results on the abstract must be revised. Some values are different from those showed on the results section. Also, there is a sentence that is not correct. Please, revise.

Reviewer #2: This is a very interesting and well done study, and the rare paper for which I have almost no additional suggestions to make! The study is sound and contributes to the literature on data sharing and the effect of data availability requirements. My only suggestion would be that it might be interesting to discuss the number of papers that are NOT open despite being published in a journal that requires open data (22 out of 55 according to Table 2). Were these journals that just encouraged rather than required open data, or were papers published despite not following the policy? I would be very interested to know how almost half the papers in this category ended up not having open data.

6. PLOS authors have the option to publish the peer review history of their article (what does this mean?). If published, this will include your full peer review and any attached files.

Reviewer #1: No

Reviewer #2: **Yes: **Lisa Federer

---

## [Author Response · Author response to Decision Letter 0]

26 Feb 2021

We thank the reviewers for their useful feedback. which has definitely improved the quality of our manuscript. Please see the "Response to Reviewers" document for a detailed response to each point raised.

---

## [Editor Report · Decision Letter 1]

22 Mar 2021

PONE-D-20-29718R1

A descriptive analysis of the data availability statements accompanying medRxiv preprints and a comparison with their published counterparts

PLOS ONE

Dear Dr. McGuinness,

Thank you for submitting your manuscript to PLOS ONE. After careful consideration, we feel that it has merit but does not fully meet PLOS ONE’s publication criteria as it currently stands. Therefore, we invite you to submit a revised version of the manuscript that addresses the points raised during the review process.

ACADEMIC EDITOR:

Thank you for revising the manuscript. I have only two minor comments:

It is not clear how the data extraction related to the journals' data-sharing policies was performed. Please, clarify that.

The conclusion should be aligned with the objectives and results. Please, revise that.

We look forward to receiving your revised manuscript.

Kind regards,

Rafael Sarkis-Onofre

Academic Editor

PLOS ONE

Journal Requirements:

Additional Editor Comments (if provided):

Academic editor:

Thank you for revising the manuscript. I have only two minor comments:

It is not clear how the data extraction related to the journals' data-sharing policies was performed. Please, clarify that.

The conclusion should be aligned with the objectives and results. Please, revise that.

---

## [Author Response · Author response to Decision Letter 1]

13 Apr 2021

A detailed response to the editorial comments raised are contained in the "Response to Reviewers" document.

---

## [Editor Report · Decision Letter 2]

16 Apr 2021

A descriptive analysis of the data availability statements accompanying medRxiv preprints and a comparison with their published counterparts

PONE-D-20-29718R2

Dear Dr. McGuinness,

We’re pleased to inform you that your manuscript has been judged scientifically suitable for publication and will be formally accepted for publication once it meets all outstanding technical requirements.

Kind regards,

Rafael Sarkis-Onofre

Academic Editor

PLOS ONE

Additional Editor Comments (optional):

All of my concerns were addressed.
---

## [Editor Report · Acceptance letter]

5 May 2021

PONE-D-20-29718R2 

A descriptive analysis of the data availability statements accompanying medRxiv preprints and a comparison with their published counterparts 

Dear Dr. McGuinness:

I'm pleased to inform you that your manuscript has been deemed suitable for publication in PLOS ONE. Congratulations! Your manuscript is now with our production department. 

Kind regards, 

on behalf of

Dr. Rafael Sarkis-Onofre 

Academic Editor

PLOS ONE